# Molecular Drug Discovery of Single Ginsenoside Compounds as a Potent Bruton’s Tyrosine Kinase Inhibitor

**DOI:** 10.3390/ijms21093065

**Published:** 2020-04-26

**Authors:** Keun Woo Lee, Woong Hee Lee, Baek-Soo Han, Jin Ha Lee, Eun Kyung Doo, Jeong-Hwan Kim

**Affiliations:** 1MODNBIO Inc. Digital road 34, Kolon Science Valley I, Guro-gu, Seoul 08378, Korea; leekw@modnbio.com (K.W.L.); xpress24@modnbio.com (W.H.L.); jinha1118@modnbio.com (J.H.L.); dek1455@modnbio.com (E.K.D.); 2Institute of Biotechnology, Chungnam National University, Daejeon 34134, Korea; 3Biodefense Research Center, Korea Research Institute of Bioscience & Biotechnology (KRIBB), 125 Gwahak-ro, Yuseong-gu, Daejeon 306-809, Korea; bshan@kribb.re.kr; 4Cardiovascular Research Institute, Graduate School of Medicine, Yokohama City University, 3-9 Fukuura, Kanazawa-ku, Yokohama 236-0004, Japan

**Keywords:** ginsenoside, drug screening, natural products, Bruton’s tyrosine kinase (BTK), BTK inhibitor, molecular docking, molecular therapeutics

## Abstract

Bruton’s tyrosine kinase (BTK) is known as a direct regulator of inflammasome, which is an intracellular target to therapeutically modulate innate immunity. Although there is great interest in developing small molecule-based drugs with BTK inhibition, there are only a few drugs available in the market, due to the difficulty of drug discovery and the potential side effects. To select suitable drug compounds to inhibit BTK signaling, molecular drug screening bioassay processes of single ginsenosides integrated with in silico molecular simulation were performed. The experimental results for the ginsenoside compositions (Rb2 and Rb3) exhibited showed that they effectively suppressed the activity of BTK expression in a rational agreement with molecular docking calculations of the compounds against the BTK binding site. They implemented a possible inhibiting effect of BTK signaling through increasing their molecular affinity for targeting BTK, enabling them to be useful in treating BTK-mediated diseases.

## 1. Introduction

Bruton’s tyrosine kinase (BTK) is a non-receptor tyrosine kinase that plays a critical role in B-cell development, differentiation, and signaling through the induction of a signal transduction that is essential for cell survival and adaptive immunity [1,2,3,4,5]. It is activated through B-cell signaling by B-cell receptors (BCR) that regulate various major signaling pathways within cells. 

However, when BTK is overexpressed by the abnormal signal transmission of BCR in B cells, the BCR signaling system undergoes excessive phosphorylation gradually and causes abnormal B-cell proliferation and pathological autoantibody formation, resulting in systemic erythematous lupus, cancers, rheumatoid arthritis, autoimmune diseases, B-cell malignancies, and inflammatory diseases [4,5]. In the proliferation of abnormal B-cells, when BTK is inhibited, signal transmission by the BCR is blocked. Thereby, the use of BTK inhibitors can be a valuable approach in the treatment of B-cell-mediated diseases [5]. BTK inhibitors have been noted as treatment drugs for these diseases, exhibiting an outstandingly effective therapeutic effect that is increased through BTK inhibition in an experimental animal model for autoimmune disease or B-cell malignancy [5]. 

Currently, many pharmaceutical companies are struggling to develop BTK inhibitors. However, due to the difficulty of development and the potential side effects, there are only three approved drugs that inhibit BTK: Ibrutinib, Acalabrutinib, and Zanubrutinib. This suggests that the development of new drugs will have an incredible economic and industrial impact. More recently, BTK was recognized as a direct regulator of a major innate inflammatory machinery, the NLRP3 (NOD-, LRR- and pyrin domain-containing protein 3) inflammasome, which is an intracellular target for therapeutically modulating innate immunity [4].

Ginsenosides are dammarane-type triterpene saponins derived from ginseng, which have been used for a long time as a preventive supplement or health functional food to improve immunity and vitality [6]. To date, research on the synthetic pathways of ginsenosides and related genetic studies is insufficient, while research on genomic proteomics has been conducted with native plants. As an active ingredient derived from natural products, ginseng ginsenoside possesses excellent therapeutic effectiveness with a high stability and little biotoxicity. Thus, numerous studies are being performed to exploit various treatments for diseases such as cancer, diabetes, heart disease, and many more [7,8,9].

Ginseng ginsenosides are classified into three types according to the difference in their structure (location and quantity of sugar moiety) (Figure 1): protopanaxadiol (PPD) group (e.g., Rb1, Rb2, Rb3, Rc, Rd, Rg3, and Rh2), protopanaxatriol (PPT) group (e.g., Re, Rg1, Rg2, and Rh1), and oleanolic acid group (e.g., Ro) [10,11]. However, their isolation/purification process for a high purity (>~98%) single compound unit in order to meet clinical needs is difficult. When separating well, each ginsenoside plays a different pharmacological role in many biological activities such as anticancer, anti-inflammation, antioxidation, antiaging, antifatigue, and physiological functions [10,11]. In particular, several ginsenoside single substances such as Rb1, Rb2, Rd, Re, Rg1, Rg3, Rg5, Rh1, Rh2, Rp1, and compound K inhibit the NF-κB signaling pathway to suppress the expression of pro-inflammatory cytokines such as TNF-α, IL-1β, and IL-6, and inflammatory enzymes such as inducible nitric oxide synthase (iNOS) and cyclooxygenase-2 (COX-2) [10,11]. 

In silico molecular docking is a computational method that rationally finds out the protein-ligand binding mode, which is as yet unknown, providing potential binding modes through calculations performed at the atomic level. If the binding mode is identified, it can greatly contribute to the study of protein function or the further development of new drugs that inhibit the function of disease markers [12,13]. Very recently, an in silico study of ginsenoside analogues as possible β-site amyloid precursor protein cleaving enzyme 1 (BACE1) inhibitors involved in Alzheimer’s disease was performed using molecular docking [14].

In this context, by harnessing the drug screening techniques for single ginsenoside compounds with the molecular simulation method, we attempted to select suitable ginsenoside therapeutic candidates which could induce the therapeutic effect of anti-inflammation by inhibiting BTK expression and also verify their affinity to target BTK. 

## 2. Results and Discussion

### 2.1. A Fluorescence Resonance Energy Transfer (FRET)-Based Kinase Assay for Intracellular BTK Inhibition

For the identification of the BTK inhibitory activity of ginsenoside-derived candidate substances and the screening of them to show pharmacological efficacy, the BTK-inhibitory activity of ginsenoside compounds and their in vitro BTK inhibitory activity was measured as the inhibition rate (inhibition %) of the decreased phosphorylation rate compared to the BTK activity without ginsenoside treatment in a FRET assay. Firstly, through repeated experiments, the BTK concentration when the phosphorylation rate of tyrosine causing the FRET reaction was 20%–40% corresponded to the optimal concentration of BTK, 0.625 μg/mL, which corresponded to the 30.63% phosphorylation rate (Figure 2).

Thus, the 20 candidate substances derived from ginsenoside were treated with 0.625 μg/mL of purified BTK at concentrations of 25, 50, and 100 μg/mL, and the BTK inhibition activity was confirmed by the BTK inhibition % of the candidates through repeated experiments (Figure 3). Accordingly, it was confirmed that Rh2, Rb2, Rb3, Rb5, and PPT exhibited a remarkable BTK inhibitory effect of 70% or more, depending on the concentration. The IC_50_ levels that inhibited BTK activity were 61.05 μM, 9.27 μM, 50.96 μM, 104.30 μM, and 136.35 μM, correspondingly (IC_50_ of Ibrutinib: 0.63 nM).

### 2.2. Solubility Test of Ginsenoside Compounds

In spite of the promising pharmaceutical efficacies of the ginsenoside compounds, the therapeutic potential of ginsenosides is largely restricted by their lowering bioavailability when orally distributed (< 5%) due to their undesirable physicochemical properties, such as low water solubility, low biomembrane permeability, instability in the gastrointestinal fluid, and extensive body metabolism [15]. Therefore, the solubility of the ginsenosides in an aqueous media solution was validated, as shown in Figure 4. Depending on the solubility of each candidate substance, the black text under the plate is clearly visible when dissolved well. Since UV/vis absorbance spectra-based turbidimetry has the advantage of being fast and non-destructive in common drug solubility tests, a determination of their turbidity or optical density (OD) in liquid cultures was employed to monitor the water-solubility of the ginsenoside candidates. The λmax of the UV/vis absorbance spectra of the ginsenoside compounds and their Roswell Park Memorial Institute (RPMI) medium were identical at 540 nm, and the difference in OD between the absorbances of the ginsenosides and the media, used only as a control, was plotted (Figure 4). The increased absorbance signals of the PPT and PPD were derived from the increased OD signal of turbidity-based interference by insoluble compounds in the medium. The corresponding UV/vis absorbance results of each candidate substance in the media is discussed as follows:

Rh2 in the medium was slightly soluble at 1.2 mM, showing a high IC_50_ value (61.05 μM), and the absorbance at 540 nm was increased by 0.271 compared to the control. The absorbance of Rh2 at 0.4 mM and 0.6 mM was increased by 0.037 and 0.111, respectively. Based on the above results, it was confirmed that the lower the concentration of Rh2 was, the better it was dissolved in the medium. Rb2 in the medium was transparent even at 0.2 mM, showing the lowest IC_50_ value (9.27 μM), and the absorbance at 540 nm was increased by 0.042 compared to the control group. The absorbance at 0.6 mM and 1.0 mM was decreased by 0.007 and 0.006, respectively, confirming that Rb2 is soluble in the medium even at a high concentrations. Rb3 at 0.9 mM was also transparent, showing a high IC_50_ value (50.96 μM), and the absorbance at 540 nm was increased by 0.047 compared to the control group. The absorbance of the 2.7 mM sample was reduced by 0.004 compared to the control, confirming that Rb3 is soluble in the medium even at a high concentration. Rb5 in the medium was transparent at 1.9 mM, showing a high IC_50_ value (104.30 μM), and the absorbance was increased by 0.032 compared to the control. The absorbance at 3.8 mM was reduced by 0.021 compared to the control, confirming that Rb5 is soluble in the medium even at a high concentration. The PPT at 1.7 mM was intensively turbid, showing the highest IC_50_ value (136.35 μM), and the absorbance was increased by 1.118 (over saturated OD) due to the murky solution compared to the control group. At 0.5 mM and 0.8 mM, the absorbances relating to concentrations of about 3 and 6 times the IC_50_ values were increased by 0.309 and 0.508, respectively, compared with the control group. This confirmed that the lower the PPT concentration, the better it was dissolved in the medium. PPD samples at 0.2 mM, 0.5 mM, 1.0 mM, and 1.5 mM were also turbid and showed spectra increased by 0.155, 0.442, 0.835, and 1.153, respectively, due to the turbidity-based OD interference. Overall, the compounds of Rb2, Rb3, and Rb5 were soluble in the medium at a high concentration, and Rh2 was soluble in the medium at a low concentration, while insoluble at a high concentration. Both PPT and PPD were poorly soluble in the medium at higher concentrations and were less soluble than other compounds at low concentrations, which is consistent with their hydrophobic chemical structures. Therefore, they are undesirable for use in further tests due to their poor solubility.

Considering the above results of the sample preparations to be treated in cells, the specific concentrations of the ginsenoside compounds in the cellular assays were employed in further assays. 

### 2.3. Confirmation of Intracellular Activity through Bioassays

Although BTK is almost constant in cells, the expression level of p-BTK varies depending on the degree of BTK activation and inhibition, and thus the expression level of p-BTK is based on the expression of BTK (p-BTK/BTK). By comparing the p-BTK/BTK ratio in the cells of the five ginsenoside candidates confirmed by ELISA, the five ginsenoside candidates (Rb2, Rb3, Rb5, PPT, and PPD) at a solubility-dependent concentration level exhibited substantial BTK inhibitory activity compared to that of Ibrutinib or the DMSO-treated group (Figure 5). Among them, Rb2 and Rb3 presented >23% BTK inhibitory activity, while the PPT and PPD showed 15.98% and 9.58%, respectively, at low concentrations. This indicates that the five selected candidates exerted their efficacy primarily by blocking BTK-mediated signaling in the cellular environment.

Moreover, by comparing the ratio of p-BTK/BTK based on the expression of p-BTK, BTK, and β-tubulin, the effect of inhibiting BTK against ginsenoside compounds in cells was confirmed. The BTK inhibitory efficacy of the drug candidates in Ramos cells was evaluated by Western blotting, as presented in Figure 6. Based on the p-BTK/BTK value of the stimulation (α-IgM antibody (+), DMSO) group as a reference (100%), which was the result of comparing and analyzing the p-BTK/BTK values of all the samples, while that of the non-stimulation group was found to be 63.07%. The p-BTK/BTK value of the stimulation group increased by 36.93% compared to the non-stimulation group, and the BTK confirmed the degree of the phosphorylation of BTK when stimulated by an α-IgM antibody. Both Rb2 and Rb3 exhibited notable BTK inhibitory activity (~17%) that was slightly higher than that of Ibrutinib at 0.4 nM, as well as the stimulation group. That of the PPT was about 30%, while the PPD showed about 40%, appearing to have a superior BTK inhibitory efficacy, excluding the poor solubility that requires a proper dispersion/delivery method for further usage.

Taken together, the five selected candidates exerted their efficacy primarily by blocking BTK-mediated signaling in the cellular environment. Yet, considering their solubility in water compared with the other compounds, it is assumed that Rb2 and Rb3 would be promising candidates in providing a pharmacological potential to enable the treatment of BTK-mediated diseases. Tumor growth in solid malignancies in the tumor environment might be also inhibited by the candidates. Moreover, since these compounds are derived from natural products, it is assumed that they are less toxic, less drug resistive, and have less probability of causing side effects, while the common side effects of Imbruvica include infection, anemia, neutropenia, thrombocytopenia, bruising, and cytopenia.

### 2.4. Comparison of Binding Affinity between BTK and Ginsenosides

A molecular docking simulation technique was exploited to predict the binding potential and binding mode of the ginsenosides over BTK domains, enabling the envisaging of how a compound may bind to a corresponding site when clusters form constant structures that frequently repeat at a specific site. Based on the combined form, the docking hit (genetic optimization for ligand docking (GOLD) fitness score) was calculated to estimate the bonding force that was obtained through the sum of energy, such as protein-ligand hydrogen bond energy (external H-bond), protein-ligand van der Waals (vdw) energy (external vdw), ligand internal vdw energy (internal vdw), and ligand torsional strain energy (internal torsion).

In general, the higher the docking score, the more energetic the interaction between the protein and the ligand is, representing that the compound fits well into the binding site within the protein. However, as the size of the calculated ligand increases, the number (degree of freedom) of the rotatable bonds increases, and thus it is difficult to effectively use this for conformation compared to relatively small ligands, reducing the accuracy of the docking result.

The docking result of each ginsenoside was classified into one cluster of similar binding modes repeatedly appearing among 50 binding modes. Among them, the binding mode with the highest GOLD fitness score was selected as a representative (red colored), as demonstrated in Figure 7. The range of docking scores of the selected binding modes ranged from 3 to 74, and among the results, the ginsenosides with a GOLD fitness score of over 60 were Ra1, Rb1, Rb3, Rd, Rg3 20 (R), and compound O of the PPD type. Of the PPT type, there were Rf, Rg1, and Rg2 20 (R). Rg5 in the third type and Rk1 in the fourth type exceeded 60 points. Unfortunately, there is no method to uniquely determine the final bond structure in molecular docking studies. The two most common methods are to select the structure with the highest frequency (cluster number), and another reasonable alternative is to select the structure with the highest molecular docking score (the structure with the highest binding energy).

The best way is to have an interpretation that matches the experimental results either way. Of these, the docking scores of Rb2 and Rb3 are 55.11 (cluster 1) and 68.90 (cluster 2), respectively, which are in good agreement with the selected final structures in the experimental section.

### 2.5. Structural Analysis of Binding Mode between BTK and Ginsenosides

The ginsenoside Ra1 exhibited the highest docking score of 73.92 in cluster 2, containing 7 out of 50 bonding modes. As shown in Figure 8A, the Glc6-Arap4-Xyl group of the R2 substituent of Ra1 faces the activation loop and forms a hydrogen bond with Q412, R525, N526, and N603. The Glc2-Glc group of the R1 substituent is exposed to the water environment at the end of the ATP-binding site. In the case of the ginsenoside Rb3, it contains 5 out of 50 binding modes, with a docking score of 68.90 in the cluster 2, as shown in Figure 8B. The binding mode behaved similarly to Ra1 on the whole, and the Glc6-Xyl group of the R2 substituent of Ra1 was placed ahead of the activation loop, forming hydrogen bonds with Q412, K430, and D539.

The results indicate that these 3D-docking models with potential predictive ability could prospectively be used in diverse ligand structure modification and optimization. Moreover, to better understand the molecular kinetic interaction, biophysical experiments to determine thermodynamic parameters such as the binding affinity (K_d_) of various BTK constructs binding to Rb2 or Rb3 are required; this is under investigation in our group. 

Based on the combination of the bioassay experiments and computer-aided simulation results, the molecular inhibition mechanism of the BTK signaling pathway by a ginsenoside molecule is illustrated as shown in Figure 9. 

## 3. Materials and Methods

### 3.1. Cell Culture

Ramos cells (American Type Culture Collection, Manassas, VA, USA) and the human Burkitt’s lymphoma cell line were cultured in suspension by Roswell Park Memorial Institute (RPMI) 1640 medium (Gibco, Rockville, MD, USA) with 10% fetal bovine serum (Gibco) at 37 °C in 5% CO_2_ in air. The in vitro-treated ginsenoside compounds (> ~99 %) were obtained from MODNBIO Inc. (Seoul, Korea). The Ibrutinib (PCI-32765) (Selleckchem, Houston, TX, USA) was used as a control. 

### 3.2. Western Blotting 

The Ramos cells (~3 × 10^6^) were famished of serum for 1 h and treated with ginsenosides for 1 h, and then washed with phosphate-buffered saline (Welgene, Korea), followed by stimulation with 1 μg/mL of goat F(ab’)2 anti-human IgM (Southern Biotech, Birmingham, AL, USA) for 10 min on ice. The cells were then lysed in a radioimmunoprecipitation assay (RIPA) buffer (Sigma-Aldrich, St. Louis, MO, USA) and the proteins were separated by SDS-PAGE (Bio-Rad Laboratories, Hercules, CA, USA) and transferred to polyvinylidene fluoride membranes (EMD Millipore, Billerica, MA, USA). The proteins were detected with anti-BTK Y223 (Cell Signaling Technology, Danvers, MA, USA), anti-BTK (53/BTK, Cell Signaling Technology) and anti-glyceraldehyde 3-phosphate dehydrogenase (anti-GAPDH) (Santa Cruz Biotechnology, Santa Cruz, CA, USA) antibodies. The protein signals were monitored by chemiluminescence using Enhanced ChemiLuminescence (ECL) western blotting detection reagent (EMD Millipore), and the scanned blots were quantified using a Multi-Image analyzer (Fujifilm, Japan). The phosphorylation % of each lane was determined using Multi-Gauge software (Fujifilm). The IC_50_ value was measured using control α-IgM+ set at 100% and control α-IgM− set at 0%.

### 3.3. Solubility Characterization of Ginsenoside Compounds

To determine the concentration of ginsenoside compounds to be treated on the cells, the solubility of each compound in RPMI-1640 medium was evaluated. Specifically, the solubility of each ginsenoside compound was observed by UV/vis absorbance at 540 nm, and the transparency of the medium was measured by the visually observed degree to which the marked black text at the bottom of the plate was visible (transparency of the medium). In addition, the change in absorbance at 540 nm compared to the control was measured by using the medium not treated with the ginsenoside compound as a control.

### 3.4. A Fluorescence Resonance Energy Transfer (FRET)-Based Kinase Assay for Intracellular BTK Inhibition

In order to evaluate the BTK inhibitory activity of the ginsenoside compound, a kinase assay using a Z’-LYTE Kinase Assay Kit from Thermo Fisher Scientific was performed. A fluorescent microplate reader was used to measure the Fluorescence Resonance Energy Transfer (FRET) between coumarin and fluorescein dyes. The assay was optimized for BTK, as described in the manufacturer’s protocol. The BTK activity (%) was evaluated by measuring the ratio of the emission signal (445 nm/520 nm) according to the phosphorylation % of the substrate by BTK. 

When BTK is activated or inhibited, the phosphorylation % of BTK is increased or decreased accordingly, which can be confirmed by the amount of phosphorylated BTK in the cell. After treating the ginsenoside compound on the cells, the amount of BTK inhibition of the ginsenoside compound was compared and analyzed by the amount of phosphorylated BTK in the cell. Then, the cells were cultured with a ginsenoside compound at a concentration higher than the value using a biochemical kit, and the absorbance was measured at 450 nm through a cell-based phosphorylation enzyme-linked immunosorbent assay (ELISA) to measure the phospho-BTK (p-BTK). The BTK to GAPDH expression ratio was confirmed.

### 3.5. Confirmation of Intracellular Activity through ELISA Assay

The Ramos cells (~1 × 10^4^) were cultured in 96-well cell culture plates. After adding the compounds onto the cells, the absorbance was measured at 450 nm through a cell-based phosphorylation ELISA to confirm the expression of p-BTK, BTK, and GAPDH. All ginsenoside compounds, including a positive control group (Ibrutinib), were dissolved in DMSO. Free DMSO was used as a negative control group. The expression amount of p-BTK was compared and analyzed based on the expression of BTK and GAPDH to confirm the BTK inhibitory efficacy of the ginsenoside compounds in cells. The expression level of each p-BTK and the expression level of BTK were compared and analyzed using the ratios (p-BTK/GAPDH and BTK/GAPDH) based on the expression level of each β-tubulin. Namely, the p-BTK/BTK ratio means the (p-BTK/GAPDH)/(BTK/GAPDH) ratio, which is simply expressed as p-BTK/BTK.

### 3.6. Confirmation of Intracellular Activity through Western Blotting

The Ramos cells (~3 × 10^6^) were cultured in 6-well cell culture plates. Both the control and ginsenoside compounds were dissolved in DMSO. For the Western blotting, the α-IgM- groups and α-IgM+ groups in the presence of DMSO were used as the non-stimulation control and stimulation control, respectively, following by the quantitation of the expression ratio of p-BTK, BTK, and β-tubulin. The expression level of each p-BTK and the expression level of BTK were compared and analyzed using the ratios (p-BTK/β-tubulin and BTK/β-tubulin) based on the expression level of each β-tubulin. Namely, the p-BTK/BTK ratio means the (p-BTK/β-tubulin)/(BTK/β-tubulin) ratio, which is simply expressed as p-BTK/BTK.

### 3.7. Molecular Docking Simulation

To perform molecular docking calculations between a ginsenoside compound and a BTK molecule, a sequential processing of precise three-dimensional conformation is required (Figure 10). By comparing the kinase domain structures of about 40 BTK known to date, no region was missed. One of the best resolution structures, PDB ID: 5P9F, was selected for the molecular docking calculation. The protonation state of the prepared protein was set for the pH 7 environment using the Clean Protein protocol of the Discovery Studio (DS) program. Thirty-eight ginsenoside structures were generated using the DS, and the structure optimization was performed using the minimization protocol. The smart minimizer algorithm, max steps 5000, and RMS gradient 0.001 were applied, and the generalized born with molecular volume (GBMV) was used as the implicit solvent model. The rest of the parameter values were set as default. The molecular docking calculation was performed using the genetic optimization for ligand docking (GOLD) program. The 20 Å radius was set as a docking site based on the mass center (COM) of the co-crystal inhibitor present in the ATP binding site of the BTK’s kinase domain. Fifty poses for each compound were predicted to be sorted based on the GOLD fitness score, and the remaining parameters were designated as default.

## 4. Conclusions

Despite the numerous proven works on the pharmaceutical research into diverse types of ginsenosides derived from natural products, there are few reports on systematic methods of molecular drug screening for single ginsenosides to target a specific disease marker, BTK. In particular, the accumulated data are still insufficient for combined molecular bioassays and simulation modelling. Consequently, in this study, a high-throughput drug screening method for enhancing the disease selectivity of diverse ginsenosides was demonstrated, accompanied by a reliable, integrative approach with experimental biomolecular assays and molecular simulation techniques. Specially, the ginsenosides Rb2 and Rb3 presented the effective targetability of BTK, involving the possible therapeutic effects of anti-inflammation by inhibiting the BTK signaling through increasing the molecular affinity to target proteins such as NLRP3 inflammasome and BTK. Given the promising therapeutic potential of Rb2 and Rb3, further studies including in vivo systems are anticipated to reveal a detailed mode of action in multiple signaling pathways that is different from existing treatments. This will maximize the clinical potential of ginsenosides as a molecular target-specific natural medicine, thereby further benefiting global health.

## Figures and Tables

**Figure 1 ijms-21-03065-f001:**
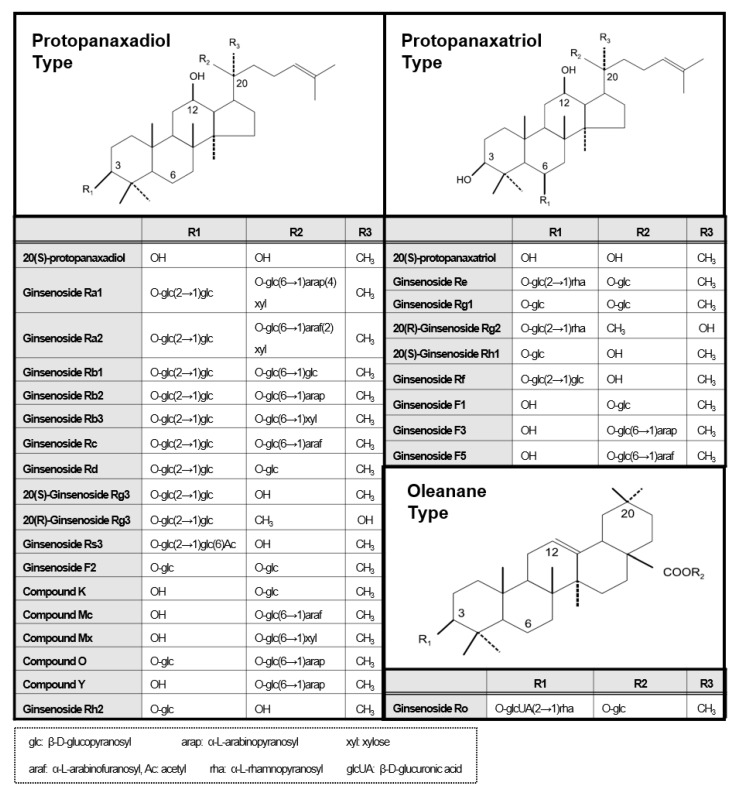
Diverse molecular structures of ginsenoside compounds.

**Figure 2 ijms-21-03065-f002:**
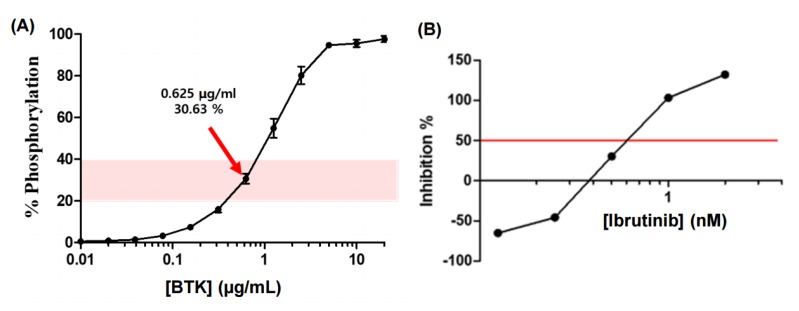
Quantitation of purified Bruton’s tyrosine kinase (BTK) concentration by a fluorescence resonance energy transfer (FRET)-based phosphorylation assay, presenting the BTK inhibition activity of the ginsenoside compounds (**A**). The concentration of Ibrutinib that inhibits 50% of BTK (IC_50_) was confirmed as 0.63 nM (**B**).

**Figure 3 ijms-21-03065-f003:**
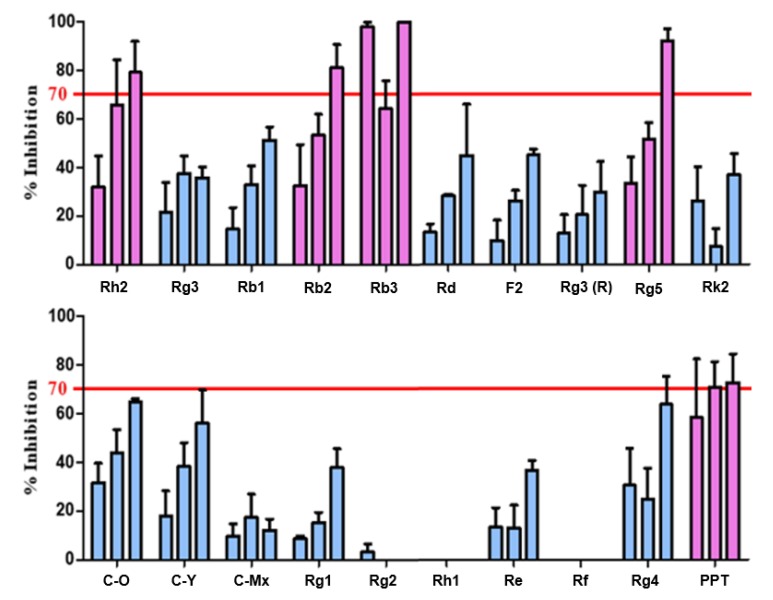
The initial drug screening of the various candidate drug substances derived from ginsenoside using their BTK inhibition activity measurement. Five drugs (pink colored) were screened out of 20 candidate substances and used for further screening steps.

**Figure 4 ijms-21-03065-f004:**
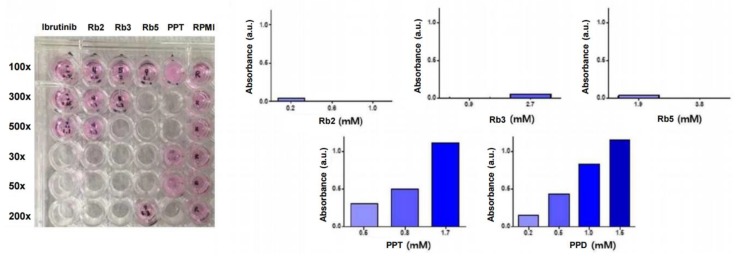
Solubility characterization of ginsenoside compounds (Rb2, Rb3, Rb5, PPT, PPD) in the culture medium. After dissolving 5 candidate substances in the Roswell Park Memorial Institute (RPMI) media, the transparency of the dissolved media is presented on microplates (the left). After dissolving each material in the media, the difference between the absorbance measured at 540 nm and the absorbance of the media, used only as a control, is presented (the right).

**Figure 5 ijms-21-03065-f005:**
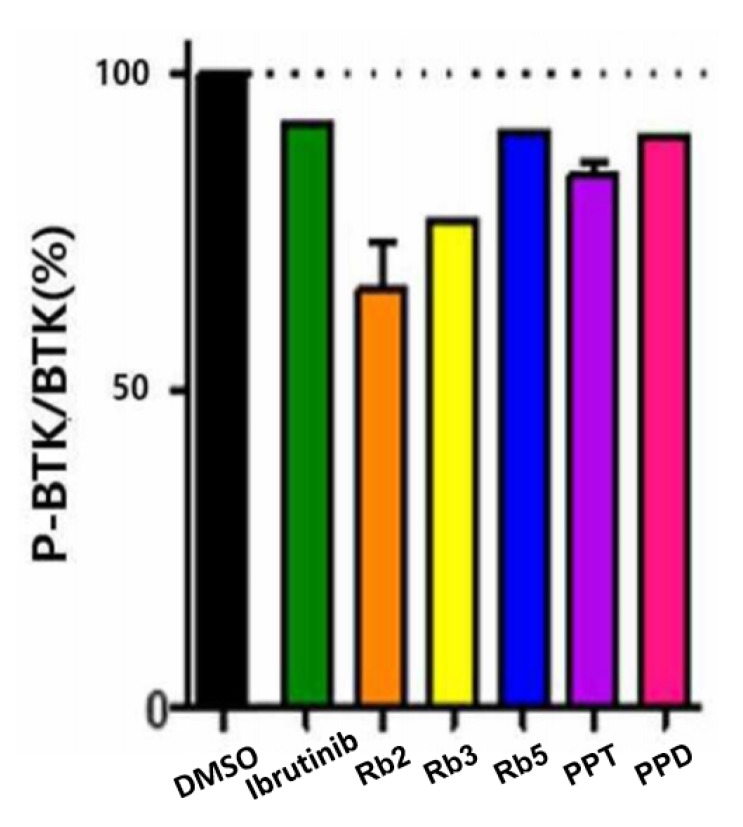
ELISA-based comparison of the BTK inhibition activity of each ginsenoside candidate treated in Ramos cells.

**Figure 6 ijms-21-03065-f006:**
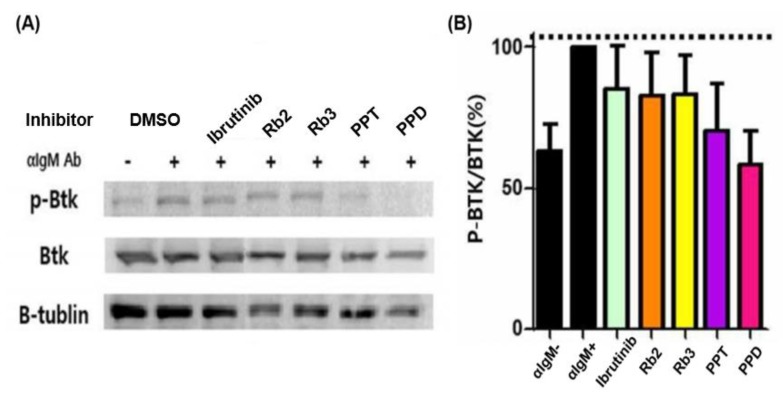
Western blotting-based comparison of BTK inhibition of each ginsenoside candidate treated in Ramos cells (**A**); quantitating the intracellular BTK inhibitory activity of the compounds by p-BTK/BTK (%) (**B**).

**Figure 7 ijms-21-03065-f007:**
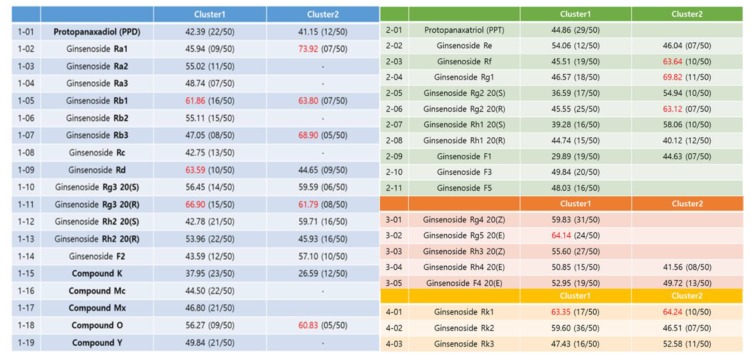
Molecular docking results (genetic optimization for ligand docking (GOLD) fitness score) of ginsenosides on the binding pocket of a BTK domain.

**Figure 8 ijms-21-03065-f008:**
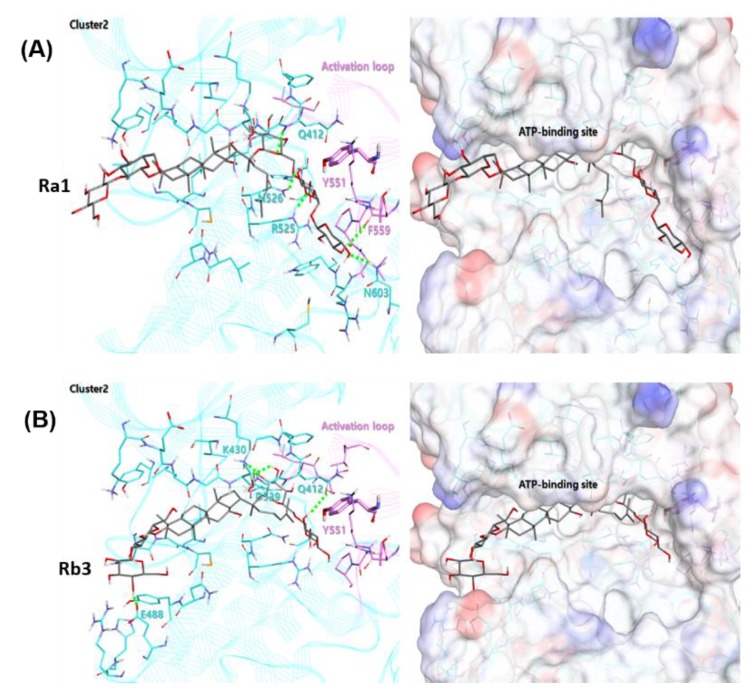
Combined docking mode of ginsenoside Ra1 (**A**) and Rb3 (**B**) in cluster 2 within a BTK domain. The hydrogen bond between the ginsenoside and BTK is indicated by a light green dotted line, and all the amino acid residues that contact the ginsenosides are light blue. The activation loop includes the key amino acid residue Y551, and the residues corresponding to the loop are shown in pink.

**Figure 9 ijms-21-03065-f009:**
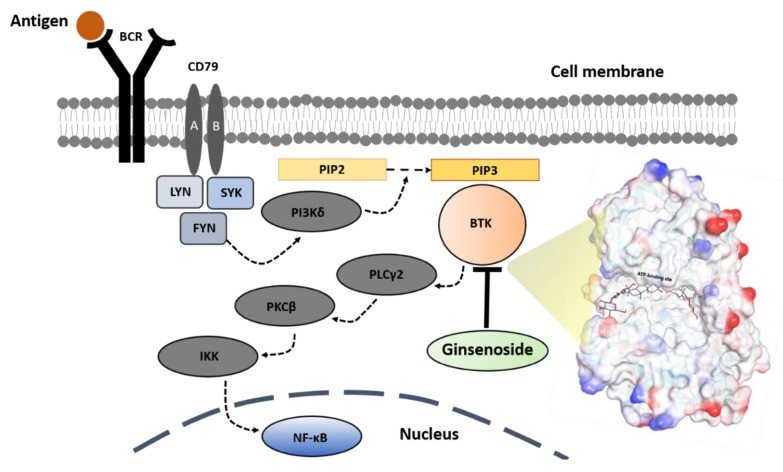
A proposed mechanism of antigen-dependent B-cell receptor (BCR) signal transduction [16] and its targeting by a small molecule-based inhibitor (ginsenoside Rb3). Antigen binding to the BCR initiates kinase-mediated signal transduction and the aggregation of the BCR with its co-receptors (CD79A and CD79B), which become phosphorylated by the recruited tyrosine kinases, Lck/Yes novel tyrosine kinase (LYN) and spleen tyrosine kinase (SYK). SYK activates phosphoinositide 3-kinase (PI3Kδ), which converts phosphatidylinositol 4,5-bisphosphate (PIP2) to phosphatidylinositol 3,4,5-triphosphate PIP3. PIP3 serves as a docking site for the cytoplasmic kinase, Bruton’s tyrosine kinase (BTK). Thereby, BTK phosphorylates and activates phospholipase C gamma 2 (PLC2γ), which produces a set of second messengers to activate protein kinase C beta (PKCβ). PKCβ phosphorylates IκB kinase (IKK) to activate nuclear factor B (NF-κB) transcription factors that regulate the gene expression necessary for B cell survival and proliferation. Through the bioassay and molecular docking result, the Rb3 molecule was selected as a novel and highly potent inhibitor of the BTK enzyme.

**Figure 10 ijms-21-03065-f010:**
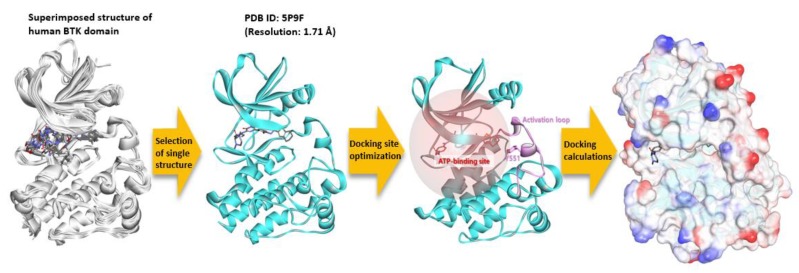
Molecular simulation process of a ginsenoside molecule’s docking site optimization in the ATP binding site within a BTK domain and the fitness calculations.

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
