# Peer review of "Molecular Drug Discovery of Single Ginsenoside Compounds as a Potent Bruton’s Tyrosine Kinase Inhibitor"

_ijms, 2020, doi:10.3390/ijms21093065_

Round 1

Reviewer 1 Report

Some remarks

1/ The results on the solubility of the compounds are not well presented and might be rewritten a bit. Why are some compounds like PPT better soluble at low concentration even if the absorbance increases with the concentration? (following the Lambert-Beer law) I imagine there can be saturation. the transparancy plots tell this?

2/ It is well known that docking scores often do not fit well with measured activities. The docking experiments can give a hint how the compounds could bind to the enzyme and which interactions might be involved. Therefore, the sentence (line 241-242) looks a bit wishful thinking to me.

Author Response

Reviewer 1

1/ The results on the solubility of the compounds are not well presented and might be rewritten a bit. Why are some compounds like PPT better soluble at low concentration even if the absorbance increases with the concentration? (following the Lambert-Beer law) I imagine there can be saturation. the transparency plots tell this?

The Authors’ Response: We would like to thank the reviewer for his/her kind comments.

The paragraph for the solubility has been rewritten by fixing some error.

The review’s comment on the optical characterization of solubility has been addressed and highlighted (pp.5, Line 113-120) in the manuscript as below:

“Since UV/vis absorbance spectra-based turbidimetry has the advantage of being fast and non-destructive in common drug solubility test, the determination of the turbidity or optical density (OD) in liquid cultures was employed to monitor the water-solubility of ginsenoside candidates. The λmax of UV/vis absorbance spectra of the ginsenoside compounds and the RPMI medium is identical as 540 nm, and the difference of OD between the absorbances of ginsenosides and media only as a control was plotted (Figure 4).The increased absorbance signals of PPT and PPD were driven from the increased OD signal of a turbidity-based interference by the insoluble compounds in the medium.”

2/ It is well known that docking scores often do not fit well with measured activities. The docking experiments can give a hint how the compounds could bind to the enzyme and which interactions might be involved. Therefore, the sentence (line 241-242) looks a bit wishful thinking to me.

The Authors’ Response: We would like to thank the reviewer for his/her kind comments.

The review’s comment on the sentence has been rewritten and highlighted in the manuscript (pp. 10, line 321-322) as below:

“Through the bioassay and molecular docking result, Rb3 molecule was selected for a novel and high potent inhibitor of BTK enzyme.”

Reviewer 2 Report

Molecular Drug Discovery of Single Ginsenoside Compounds as A Potent Bruton's Tyrosine Kinase Inhibitor

ijms-777409

Comments : 

it is very interesting to see bioassay and modelling data correlate well and nicely presented, at least a few of the interesting compounds binding affinity should be demonstrated by biophysical experiments. 

Author Response

Reviewer 2

it is very interesting to see bioassay and modelling data correlate well and nicely presented, at least a few of the interesting compounds binding affinity should be demonstrated by biophysical experiments. 

The Authors’ Response: We thank the reviewer for the valuable comment.

As we agreeing with the reviewer’s kind feedback, the experimental work to determine thermodynamic experimental parameters such as the binding affinity (Kd or Ki) is also worth spending time on to ensure the clue on the biomolecular interaction, which allows for further optimization of the ginsenoside compounds. However, the study requires a series of experiments using time & cost-consuming intensive biophysical techniques such as isothermal titration calorimetry (ITC) or surface resonance plasmon (SPR). Although the additional work seems out of scope in this report, it is currently being performed in our research group as a following-up study of this paper.

In response to the reviewer’s comment, a sentence has been added in pp. 9, Line 301-304 (highlighted) in the manuscript as below:

“Moreover, to better understand the molecular kinetic interaction, biophysical experiments to determine thermodynamic parameters such as the binding affinity (Kd) and stoichiometry for various BTK constructs binding to Rb2 or Rb3 should be required, which is under investigated in our group.”

Round 2

Reviewer 1 Report

no more comments

Reviewer 2 Report

no more revisions needed